# Applying Machine Learning for Analyzing Shooting Importance in Modern Pentathlon

Jieung Kim [1] and Jongchul Park [2,3,*]

1 Department of Sport Science, Korea Institute of Sport Science, Seoul 05540, Republic of Korea
2 Department of Marine Sports, Pukyong National University, Busan 48513, Republic of Korea
3 Marine Designeering Education Research Group, Pukyong National University, Busan 48513, Republic of Korea
* Correspondence: jcpark@pknu.ac.kr

**Abstract**

This study aimed to analyze and quantify the relative importance of each shooting series (Series 1 through 4) to variations in total shooting performance during the laser run event in Modern Pentathlon using machine learning (ML) models and interpretability methods. Individual shooting times ($n = 1453$) were collected from six international competitions hosted by UIPM in 2024, and 2-ML models, Random Forest and XGBoost, were trained to predict total shooting time. To interpret model results, two interpretability methods—Permutation Importance and SHAP (SHapley Additive exPlanations)—were applied to the trained models, and their performance was evaluated using Mean Absolute Error (MAE) and R-squared ($R^2$) with 5-fold cross-validation. Series 4 shooting consistently exhibited the highest importance in explaining variations in total shooting time for both male and female athletes, with the XGBoost model interpreted using SHAP achieving the highest predictive accuracy ($R^2 = 0.97$). Additionally, Permutation Importance identified Series 1 as particularly influential among male athletes, addressing potential biases in Random Forest's Mean Decrease Impurity measure. These findings highlight the importance of managing concentration, rapid recovery, and psychological pressure during the final shooting stage, alongside the significance of stability in earlier series, providing actionable, data-driven strategies for training and psychological preparation in Modern Pentathlon athletes.

**Keywords:** modern pentathlon; shooting; random forest; XGBoost; permutation importance; SHAP

## 1. Introduction

The field of sports science is undergoing a transformative shift driven by the integration of advanced data analytics and machine learning (ML), enabling more sophisticated performance evaluation and personalized training strategies across various disciplines [1]. In time-sensitive sports such as Modern Pentathlon, where marginal differences in performance can determine victory or defeat, precise and data-driven analysis of performance variables is crucial [2].

Modern Pentathlon is a multidisciplinary sport composed of fencing, swimming, equestrian, shooting, and running. Among these, the laser run, introduced at the 2012 London Olympics, has emerged as a decisive final event, combining laser pistol shooting with cross-country running in alternating sequences [3]. This unique structure demands high aerobic capacity, rapid recovery, and sustained psychological composure, particularly

because athletes must alternate repeatedly between intense physical exertion and precise shooting [4]. Laser runs, which are played under extreme fatigue late in the game, require a high level of concentration and skill from each player, and are a key factor in determining rankings [5]. Accordingly, optimizing shooting performance has become a focal point for coaches and sports scientists [5].

While previous research in Modern Pentathlon has focused primarily on physiological metrics and general performance statistics [5,6], there remains a significant gap in quantitative, data-driven analysis of shooting performance using real-world competition data. Specifically, the laser run includes four consecutive shooting series (Series 1–4), and although total shooting time is the sum of these four segments, the relative contribution of each series to the variation in total time and, by extension, to performance outcomes has not been systematically investigated. Basic analytical approaches, such as linear regression or correlation analysis, may appear adequate for assessing proportional effects; however, these techniques often fall short when applied to complex, nonlinear, and interdependent data arising from actual competitive settings [7].

Previous research in sports data analysis has predominantly focused on statistical methodologies or the application of single machine learning models, such as correlation and regression analyses [8]. However, such methodologies frequently prove inadequate in capturing complex, nonlinear relationships and intricate interactions among various features. Furthermore, when predictive models operate as "black boxes," discerning which features significantly influenced a given prediction becomes challenging [9]. This inherent lack of interpretability can undermine confidence in the model and complicate data-driven decision-making within authentic training environments. For instance, the Mean Decrease in Impurity (MDI), a feature importance metric commonly employed in tree-based models such as Random Forest, albeit computationally efficient, can exhibit bias when applied to highly correlated features or high-dimensional categorical variables. As a result, it may inaccurately represent the true contribution of features to the model's generalization performance [10]. This issue is particularly pronounced when analyzing temporally dependent variables, such as sequential shooting times, where strong correlations between adjacent series can lead MDI to undervalue or disproportionately emphasize certain features [10].

The aim of this study is to apply machine learning (ML)-based feature importance methodologies—mean reduced impurity (MDI), Permutation Importance, and Shapley additive explanations (SHAP)—to laser run shooting data obtained from international Modern Pentathlon competitions. The main objective is not to simply predict the total shooting time based on the sum of the 4-shooting series (Series 1–Series 4), but to quantitatively analyze the relative contribution of each series, considering the complex interdependencies and nonlinear relationships that can occur in real competitive conditions. We hope to provide a practical basis for the athlete's training and competition strategy.

First, Random Forest's internal MDI scores are used to capture the internal structure of a tree-based model by measuring the extent to which each feature reduces node impurity during decision-making [11]. This metric is a quick and intuitive way to determine importance. It provides insights into the intrinsic relevance of features within the model's decision-making process.

Second, Permutation Importance is a powerful, model-agnostic technique used to quantify the actual contribution of each feature to a model's generalization performance. It works by measuring how much the model's performance degrades when the values of a specific feature are randomly shuffled. This method serves as a valuable complement to other feature importance techniques, such as Mean Decrease Impurity (MDI), because it offers a more objective metric of feature relevance. Permutation Importance is less af-

fected by correlations between variables and can effectively assess generalized importance, helping to mitigate potential biases [12].

Third, XGBoost, a powerful boosting algorithm renowned for its exceptional predictive performance, is integrated with SHAP (SHapley Additive exPlanations). SHAP is a robust interpretability framework, grounded in cooperative game theory, that fairly attributes the magnitude and direction of each feature's contribution to individual predictions [13]. This is particularly valuable in sports analytics and can be applied to a variety of machine learning models, facilitating the generation of personalized feedback. For example, while Series 4 may be influential for most athletes due to fatigue accumulation, some individuals may be more affected by earlier-stage instability (e.g., Series 1 or 2). SHAP quantifies such personalized influences, which cannot be uncovered through simple correlation or linear regression techniques.

This approach enables not only global feature importance analysis but also granular, instance-level interpretation. For example, in a study predicting NBA game outcomes [14], SHAP was utilized to visualize how specific performance indicators, such as points scored, field goal percentage (FG%), and assists, influenced the win probability within distinct game contexts. Analogously, in the present study, SHAP facilitates the identification of which shooting series contributed most significantly to an athlete's prolonged shooting time, thereby enabling nuanced diagnostic insights at the individual level.

By integrating these three methodological approaches—Random Forest (employing MDI), Permutation Importance, and XGBoost with SHAP—this study provides a comprehensive analysis of the key factors influencing total shooting time. The findings are anticipated to inform evidence-based strategies for enhancing athlete performance in Modern Pentathlon shooting.

Ultimately, through comparative analysis of these three methodologies, the study aims to provide a multi-dimensional understanding of the pivotal factors influencing shooting performance in Modern Pentathlon. The results will not only guide athletes and coaches on which series to prioritize during training but also illuminate the interpretability strengths and limitations inherent to each machine learning technique. Therefore, this study proposes a new method for analyzing shooting performance by integrating explainable machine learning-based analysis techniques using official competition records, breaking away from existing limited analysis methodologies. This will enable strategic preparation for the laser run in modern Pentathlon in the future.

## 2. Materials and Methods

### 2.1. Participants

The participants of this study consist of male and female athletes who competed in a total of six international Modern Pentathlon competitions held in 2024, including World Cups 1 through 4, the World Cup Final, and the World Championships. Table 1 presents the characteristics of data collected from six international competitions in 2024 for this study, divided by competition and round. A total of 1453 athletes were included in the analysis, comprising 742 men and 711 women. The competition types within the laser run event included 822 athletes in the preliminary rounds, 422 in the semifinals, and 209 in the finals, representing 83 different countries. The thing to consider is that, due to the nature of international competitions, there may be differences and biases in performance based on differences in skills between rounds (qualifying, semifinal, and final) and individual players.

**Table 1.** Participants per competition.

| Competition | Round | Male (n) | Female (n) | Sum |
|---|---|---|---|---|
| World Cup 1 | Final | 18 | 18 | 36 |
| | Semi final | 34 | 36 | 70 |
| | Qualifications | 87 | 81 | 168 |
| World Cup 2 | Final | 17 | 18 | 35 |
| | Semi final | 36 | 35 | 71 |
| | Qualifications | 93 | 84 | 177 |
| World Cup 3 | Final | 17 | 18 | 35 |
| | Semi final | 36 | 35 | 71 |
| | Qualifications | 87 | 81 | 168 |
| World Cup 4 | Final | 17 | 16 | 33 |
| | Semi final | 35 | 36 | 71 |
| | Qualifications | 83 | 77 | 160 |
| World Cup 5 | Final | 18 | 17 | 35 |
| | Semi final | 35 | 35 | 70 |
| Championship | Final | 17 | 18 | 35 |
| | Semi final | 33 | 36 | 69 |
| | Qualifications | 79 | 70 | 149 |
| Total | | 742 | 711 | 1453 |

## 2.2. Data Collection and Research Variables

For the purpose of this study, official laser run records were collected from six international competitions organized by the Union Internationale de Pentathlon Moderne (UIPM) in the year 2024. The assembled dataset comprised shooting times for each of the four series (Series 1–4), running time, and the total laser run time.

The independent variables (predictors) were designated as the individual shooting times for each of the four series (Series 1–4), whereas the dependent variable was defined as the total shooting time (Total Series). In this study, ST refers to the series time in the figure and is expressed as ST1 to ST4. All variables were treated as continuous numerical variables. The raw data were utilized without Z-score normalization or scaling, given that tree-based algorithms are generally insensitive to feature scaling with respect to model performance.

Data collection was executed using Microsoft Excel (Microsoft, Redmond, WA, USA). Comprehensive data pre-processing procedures were applied prior to the commencement of formal analysis.

## 2.3. Data Processing and Statistical Analysis

First, descriptive statistical analysis was conducted using SPSS 25.0 (IBM Corp., Armonk, NY, USA) to examine the overall characteristics of the data, including measures such as standard deviation, minimum, and maximum values.

Second, in preparation for machine learning, the dataset was preprocessed using the Python programming language along with the Pandas library. Correlation analysis was performed to assess the relationships among the independent variables and between the independent and dependent variables. In addition, multicollinearity was examined to identify potential redundancy among predictors.

Third, the preprocessed dataset was randomly split into training and test sets using a 70:30 ratio. Model training and hyperparameter tuning were conducted using the training set, while final model evaluation and calculation of Permutation Importance were performed on the test set to validate the model's generalization performance.

Fourth, based on the data, feature importance was calculated using three machine learning algorithms—Random Forest, Permutation Importance, and XGBoost—to determine which shooting series most significantly affected the total shooting time. Shooting time for each series was defined as the number of seconds required to complete the shot, with shorter durations interpreted as faster shooting performance.

Lastly, model performance was evaluated using the Mean Absolute Error (MAE) and the coefficient of determination ($R^2$). Feature importance values from each model were extracted and visualized. To ensure reliable and robust assessment of generalization capability, how well the model performs on unseen data, a 5-k-fold cross-validation was performed.

*2.4. Machine Learning Models and Feature Importance Analysis*

In this study, Machine Learning (ML) models were employed to predict total shooting time based on the four individual shooting series (ST1–ST4). In parallel, Explainable Artificial Intelligence (XAI) techniques were utilized to interpret the contribution of each input feature to the model's output. While often used in conjunction, ML and XAI serve distinct yet complementary roles in data analysis. ML is primarily concerned with optimizing predictive accuracy by learning patterns in data, often functioning as a "black box" where the decision logic is not transparent. In contrast, XAI is dedicated to revealing and interpreting the underlying decision-making process of ML models. This dual approach ensures both performance and transparency, allowing the model to make accurate predictions while also providing insights that are interpretable and actionable for coaches, analysts, and athletes.

2.4.1. Random Forest and Mean Decrease in Impurity (MDI)

Random Forest is a robust non-parametric ensemble learning algorithm that enhances predictive accuracy by aggregating the outputs of multiple decision trees. Each tree is trained independently using a bootstrap sample of data and a random subset of features selected at each node split [15]. The final prediction is determined by averaging the predictions of all trees (for regression) or through majority voting (for classification). To reduce overfitting, only a subset of all available features is considered for splitting at each node.

Feature importance in Random Forest is quantified based on how much each feature contributes to decreasing impurity across all splits in all trees. This metric is referred to as the Mean Decrease in Impurity (MDI). In this study, MDI-based feature importance values were computed using the Random Forest Regression implementation from the scikit-learn library.

The MDI score is calculated by averaging the impurity reduction contributed by a feature across all the nodes in all the trees where that feature is used. This score is weighted by the number of samples that reach each respective node. As such, MDI provides a quantitative measure of a feature's contribution to improving model performance and enhances model interpretability by helping identify the most influential predictors.

$$MDI(X_j) = \sum_{t \in T} 1(v(t) = X_j) \cdot \Delta i(t) \cdot \frac{N_t}{N} \tag{1}$$

Mathematical Definition of Mean Decrease in Impurity:

$\sum_{t \in T}$ : The set of all nodes across all trees in the Random Forest;
$1(v(t) = X_j)$: An indicator function that equals 1 if feature $X_j$ is used for the split at node $t$, and 0 otherwise;
$\Delta i(t)$: The decrease in impurity at node $t$;

$N_t$: The number of training samples that reach node $t$;
$N$: The total number of training samples.

2.4.2. Permutation Importance

Permutation Importance is a model-agnostic method for evaluating feature importance based on the impact of individual features on the performance of a trained model. The approach involves computing a baseline score—typically measured using the test set—to assess the model's original predictive accuracy. In this study, Mean Squared Error (MSE) and R-squared ($R^2$) were used as performance metrics.

Feature importance is determined by calculating the change in performance before and after randomly shuffling the values of each feature. A larger decrease in model performance after permutation indicates greater importance of the corresponding feature. To ensure robustness and stability, multiple permutations were conducted for each independent variable. In this study, the permutation importance function from the scikit-learn library was used, with the number of permutations set to $K = 30$.

$$PI_j = \frac{1}{K}\sum_{k=1}^{K}(S^{(k)}_{permuted,j} - S_{baseline}) \tag{2}$$

Permutation Importance Calculation Formula:

$PI_j$: Permutation Importance of feature $j$;
$S_{baseline}$: Model performance scores (e.g., MSE or MAE) on the original test dataset; lower values indicate better performance;
$S^{(k)}_{permuted,j}$: Model performance score (e.g., MSE or MAE) on the test dataset with feature $j$ permuted during the $k$-th repetition;
$K$: Number of permutation repetitions.

2.4.3. XGBoost and SHAP (SHapley Additive exPlanations)

XGBoost is an ensemble learning algorithm based on gradient boosting that sequentially combines weak learners using gradient descent optimization. It is widely recognized for achieving high predictive accuracy and robustness. The algorithm incorporates several practical features such as L1 and L2 regularization to prevent overfitting, automatic handling of missing values, and support for parallel computation, making it particularly well-suited for analyzing real-world data such as those collected in sports competitions [16].

SHAP (SHapley Additive exPlanations) is essentially a model-agnostic method, meaning it can be applied to any machine learning model to provide insights into variable importance and interpret prediction results. In the present study, XGBoost was selected as an analytical model due to its structural advantages in capturing nonlinear relationships between the independent variables (Series 1–Series 4) and the dependent variable (Total Series), as well as its ability to model complex feature interactions. This pairing allowed us to maximize the depth and utility of SHAP's explanations.

SHAP (SHapley Additive exPlanations) is an interpretability framework grounded in game theory, specifically the concept of Shapley values. It offers the advantage of providing fair and mathematically consistent explanations by decomposing a model's prediction into individual feature contributions. SHAP enables not only global interpretation of average feature importance across the entire dataset but also local explanations of specific predictions at the instance level [17].

In this study, SHAP summary plots and mean absolute SHAP values were used to visually and numerically assess which of the shooting series (Series 1 to Series 4) had the greatest influence on total shooting time.

$$\mathcal{L}(\phi) = \sum_{i=1}^{n} l\left(y_i, y_i^{(t)}\right) + \sum_{k=1}^{t} \Omega(f_k) \tag{3}$$

XGBoost's objective functional formula:

$l$: Loss function (e.g., squared error);

$y_i^{(t)}$: Predicted value at the $t$-th iteration;

$\Omega(f_k)$: $\gamma^T + \frac{1}{2} + \lambda\| \omega \|^2$: Regularization term, defined as $\gamma T + (1/2)\lambda\|\omega\|^2$, which includes the penalty for model complexity;

T: Number of leaf nodes in the current tree;

$\gamma$: Penalty coefficient for the number of leaf nodes;

$\lambda$: Coefficient for the L2 regularization term on the leaf weights ($\omega$).

### 2.4.4. Hyperparameter Tuning Process

Table 2 shows the results of hyperparameter tuning for the random forest and XGBoost models used in this study. The tuning strategy employed is a grid search and 5-fold cross-validation. For the max_depth parameter, which determines the maximum depth of each decision tree, a grid search was conducted over the values [None, 5, 10, 15, 20]. The optimal result was achieved with max_depth = None, meaning that no maximum depth constraint yielded the best model performance. This suggests that allowing the trees to grow fully without limiting their depth helped capture the complex, potentially nonlinear interactions between input features and the target variable.

**Table 2.** Model hyperparameters.

| Model | Hyperparameter | Search Range | Tuning Strategy | Optimal Value |
|---|---|---|---|---|
| Random Forest | n_estimators (Number of Trees) | [50, 100, 200, 300] | Grid Search and 5-fold CV | 100 |
| | max_depth (Tree Depth) | [None, 5, 10, 15, 20] | | None |
| | min_samples_split | [2, 5, 10, 20] | | 2 |
| XGBoost | learning_rate | [0.01, 0.05, 0.1, 0.2] | Grid Search and 5-fold CV | 0.1 |
| | max_depth | [3, 5, 7, 10] | | 5 |
| | n_estimators | [100, 200, 300, 500] | | 200 |

The min_samples_split parameter, which controls the minimum number of samples required to split an internal node, was explored over the range (2, 5, 10, 20). The best performance was obtained when min_samples_split = 2, indicating that the model benefited from a high degree of flexibility in tree growth, allowing nodes to split even with small sample sizes. This is particularly useful when modeling fine-grained patterns.

## 3. Results

### 3.1. Technical Statistics

Table 3 presents the results of technical statistics on shooting time by series according to gender. For male athletes, the shortest mean shooting time was observed in Series 3 (M = 13.03 s, SD = 4.90), while Series 1 exhibited the longest mean shooting time (M = 14.10 s, SD = 6.04). The maximum shooting time reached the regulation limit of 50.00 s in all series (48.77 s in Series 3), indicating that some athletes utilized the maximum allowable time. Notably, Series 3 demonstrated the smallest standard deviation

(SD = 4.90), suggesting relatively low variability in shooting times among male athletes during this series.

**Table 3.** Shooting time by gender and series.

| Gender | Series | N | Minimum | Maximum | Mean | Standard Deviation |
|---|---|---|---|---|---|---|
| M | Series 1 | 742 | 6.33 | 50.00 | 14.0962 | 6.04063 |
| | Series 2 | 742 | 5.74 | 50.00 | 13.3402 | 5.13639 |
| | Series 3 | 742 | 6.22 | 48.77 | 13.0252 | 4.89721 |
| | Series 4 | 742 | 6.03 | 50.00 | 13.8169 | 5.67441 |
| F | Series 1 | 711 | 5.85 | 50.00 | 14.9035 | 6.26419 |
| | Series 2 | 711 | 6.31 | 50.00 | 14.3270 | 5.80771 |
| | Series 3 | 711 | 6.60 | 50.00 | 14.3619 | 6.07767 |
| | Series 4 | 711 | 6.07 | 50.00 | 14.7397 | 6.69223 |

For female athletes, the shortest mean shooting time occurred in Series 2 (M = 14.33 s, SD = 5.81), while the longest mean time was recorded in Series 1 (M = 14.90 s, SD = 6.26). Similar to the male athletes, the maximum shooting time in all series reached the upper limit of 50.00 s. Series 4 exhibited the largest standard deviation (SD = 6.69), indicating the greatest variability in shooting performance among female athletes in this series.

Overall, male athletes showed a tendency for faster shooting times compared to female athletes. Specifically, male athletes were most efficient in Series 3, whereas female athletes achieved their best performance in Series 2. The frequent occurrence of maximum shooting times (50.00 s) across all series suggests that time delays due to missed shots may have a significant impact on recorded performance outcomes. The standard deviations observed in each series reflect disparities in shooting proficiency among athletes, with the particularly high variability in Series 4 among female athletes indicating that this series may serve as a key determinant of performance differentiation.

*3.2. Correlation Analysis*

Figure 1 shows the result of visualizing the correlation coefficient between the execution time (ST 1–ST 4) and the total shooting time (Total ST) for each shooting series for male (a) and female (b) athletes, respectively. The correlation coefficient (r) of each series and the significance probability of the corresponding value were together, and all relationships showed a statistically significant level ($p < 0.001$). In the case of male athletes, the variable that showed the highest correlation with Total ST was ST 4 (r = 0.73), followed by ST 1 (r = 0.70), ST 2 (r = 0.68), and ST 3 (r = 0.65). The correlation between series is generally relatively low, ranging from 0.2 to 0.4, showing that each series is a performance element with a certain level of independence. For female athletes, all series showed relatively high correlation with Total ST, followed by ST 2 (r = 0.75), ST 4 (r = 0.74), ST 3 (r = 0.71), and ST 1 (r = 0.69). In particular, the high correlation coefficient of ST 2 suggests that for female athletes, shooting performance in the early and mid-game may have a greater impact on the overall outcome. The correlation between series also tends to be somewhat higher than that of men, indicating that performance consistency or cumulative fatigue effects between shooting series may be more prominent in female athletes.

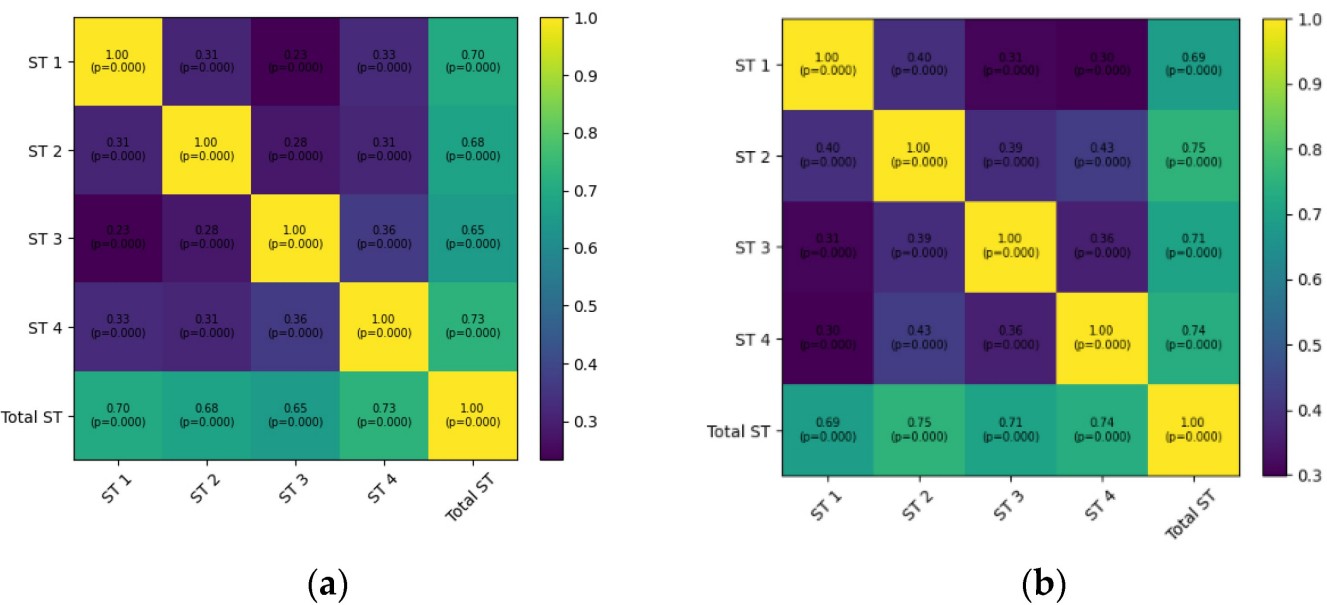

**Figure 1.** Heatmaps of correlation metrics: (**a**) Men, (**b**) Women.

*3.3. Importance by Model*

3.3.1. Mean Decrease in Impurity

Table 4 presents the importance of MDI and performance indicators of the model for each series. Figure 2 shows the results of model variable importance (Figure 2a,b) and performance evaluation (Figure 2c,d) for male and female athletes. According to the results of the Random Forest model for male athletes, the shooting time in Series 4 exhibited the highest importance in predicting total shooting time, with an MDI value of 0.3771. This was followed by Series 1 (0.2976), Series 3 (0.1824), and Series 2 (0.1430). The model's predictive performance showed a mean absolute error (MAE) of $1.927 \pm 0.411$ s and an $R^2$ of $0.937 \pm 0.027$, indicating that the model explains a high proportion of the variance in total shooting time.

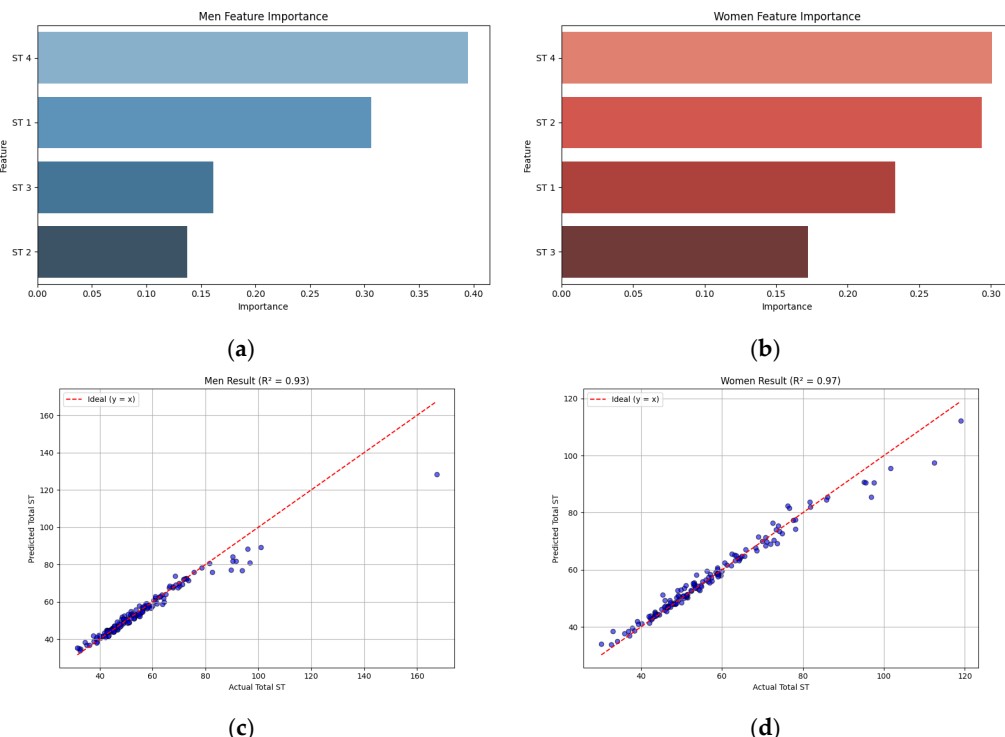

**Figure 2.** Mean Decrease in Impurity result: (**a**) Male MDI, (**b**) Female MDI, (**c**) Male $R^2$, (**d**) Female $R^2$.

**Table 4.** Series importance based on Random Forest.

| Gender | Series 1 | Series 2 | Series 3 | Series 4 | MAE | $R^2$ |
|--------|----------|----------|----------|----------|-----|-------|
| Male | 0.2976 | 0.143 | 0.1824 | 0.377 | $1.927 \pm 0.411$ | $0.937 \pm 0.027$ |
| Female | 0.2185 | 0.2789 | 0.2191 | 0.284 | $1.975 \pm 0.211$ | $0.954 \pm 0.014$ |

For female athletes, Series 4 demonstrated the highest importance, with an MDI value of 0.2835, followed closely by Series 2 (0.2789), Series 3 (0.2191), and Series 1 (0.2185). Notably, the difference in importance between Series 2 and Series 4 was minimal. The predictive performance of the model for female athletes was also high, with an MAE of $1.975 \pm 0.211$ s and an $R^2$ of $0.954 \pm 0.014$, similar to the male model.

Overall, in both male and female athletes, Series 4 was identified as the most influential factor in total shooting time based on Random Forest-derived feature importance. This finding suggests that performance in the final shooting series plays a critical role in determining total shooting performance. However, the order of importance among the other series varied by gender, potentially reflecting differences in race strategies or shooting proficiency between male and female athletes.

### 3.3.2. Random Forest and Permutation Importance

Table 5 presents the importance of permutation and the performance indicators of the model for each series. Figure 3 illustrates the results of variable importance (Figure 3a,b) and performance evaluation (Figure 3c,d) of the model for male and female athletes.

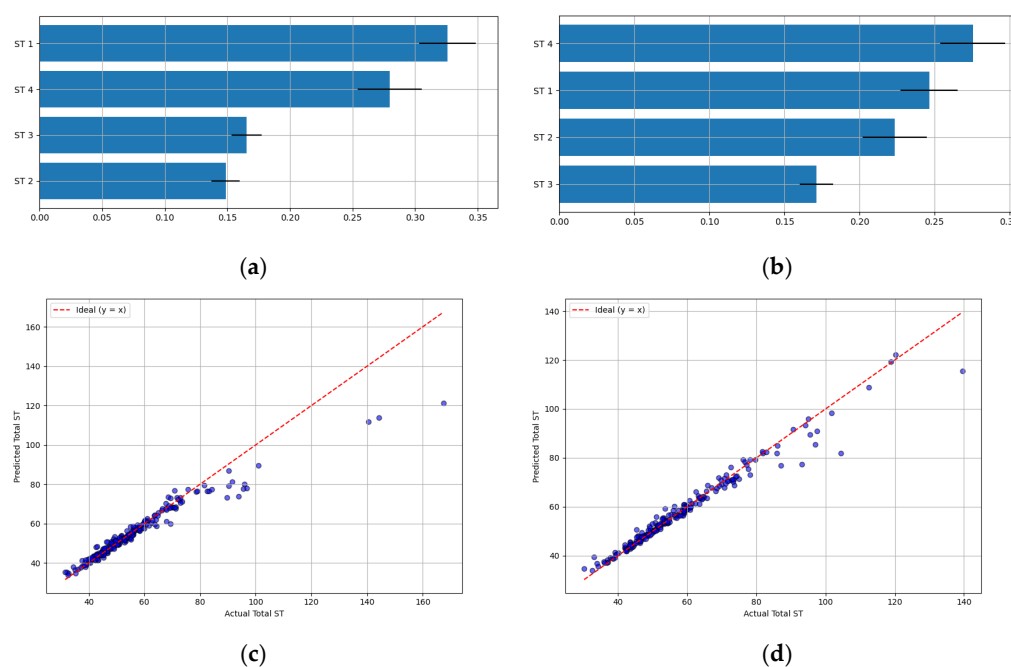

**Figure 3.** Permutation Importance result: (**a**) Male Permutation Importance, (**b**) Female Permutation Importance, (**c**) Male $R^2$, (**d**) Female $R^2$.

In the Permutation Importance analysis for male athletes, Series 1 shooting time showed the highest importance, with a score of 0.326. This indicates that random shuffling of Series 1 values led to the most significant decrease in the model's predictive performance for total shooting time. Series 2 and Series 4 each had an importance score of 0.280, while Series 3 showed the lowest importance at 0.165.

For female athletes, Series 4 shooting time exhibited the highest Permutation Importance at 0.276, followed by Series 1 (0.247), Series 2 (0.223), and Series 3 (0.171).

**Table 5.** Series importance based on Permutation Importance.

| Gender | Series 1 | Series 2 | Series 3 | Series 4 | MAE | $R^2$ |
|--------|----------|----------|----------|----------|-----|-------|
| Male | 0.326 | 0.280 | 0.165 | 0.280 | $1.927 \pm 0.411$ | $0.937 \pm 0.027$ |
| Female | 0.247 | 0.223 | 0.171 | 0.276 | $1.975 \pm 0.211$ | $0.954 \pm 0.014$ |

The results from Permutation Importance differed from those of the MDI-based feature importance derived from the Random Forest model. Notably, for male athletes, Series 4 was the most important feature according to MDI, whereas Series 1 emerged as the most influential feature based on Permutation Importance. This discrepancy can be attributed to the inherent bias in MDI, such as its tendency to distribute importance among highly correlated features, whereas Permutation Importance more directly reflects the actual contribution of each feature to the model's generalization performance.

For female athletes, both methods identified Series 4 as the most important feature, yet differences in the ranking of the remaining series were observed. These contrasts highlight the methodological distinctions between feature importance techniques and underscore the necessity of employing multiple analytical approaches rather than relying on a single metric.

### 3.3.3. XGBoost + SHAP

Table 6 presents the importance of SHAP and performance indicators of the model for each series. Figure 4 illustrates the results of XGBoost variable importance (Figure 4a,b) and performance evaluation (Figure 4c,d) for male and female athletes. In the XGBoost and SHAP analysis for male athletes, Series 4 shooting time exhibited the highest importance, with a SHAP value of 5.276, followed by Series 1 (4.673), Series 2 (3.594), and Series 3 (3.462). The predictive performance of the XGBoost model was outstanding, with a mean absolute error (MAE) of $1.834 \pm 0.384$ s and an $R^2$ of $0.970 \pm 0.025$, indicating that the model explained 97.0% of the variance in total shooting time.

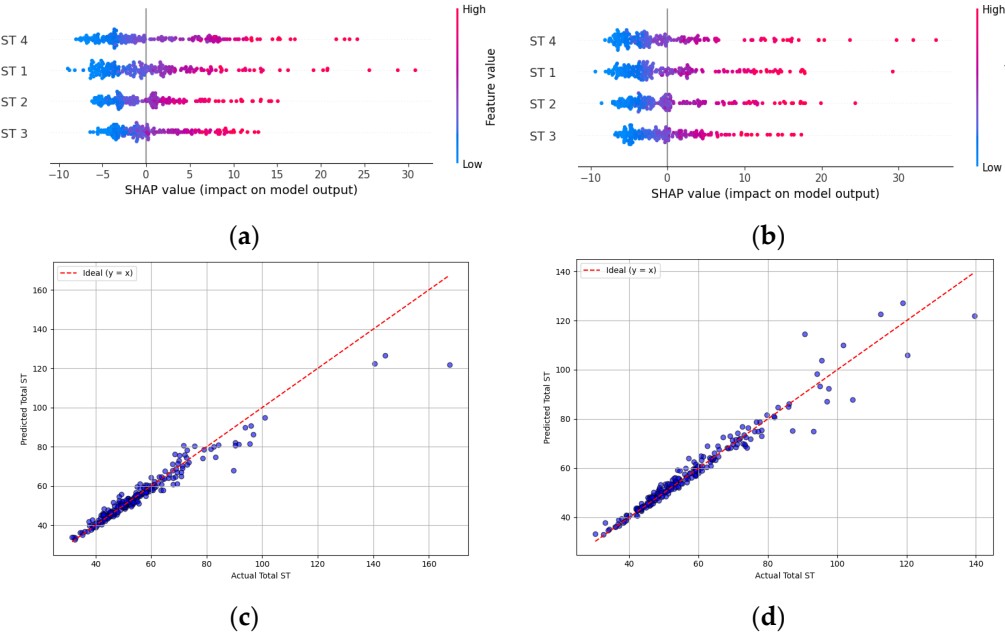

**Figure 4.** XGBoost + SHAP result: (**a**) Male SHAP, (**b**) Female SHAP, (**c**) Male $R^2$, (**d**) Female $R^2$.

**Table 6.** Series importance based on XGBoost + SHAP.

| Gender | Series 1 | Series 2 | Series 3 | Series 4 | MAE | $R^2$ |
|--------|----------|----------|----------|----------|-----|-------|
| Male | 4.673 | 3.594 | 3.462 | 5.276 | $1.834 \pm 0.384$ | $0.970 \pm 0.025$ |
| Female | 5.083 | 4.595 | 3.900 | 5.339 | $2.027 \pm 0.495$ | $0.986 \pm 0.007$ |

Similarly, for female athletes, Series 4 was also identified as the most important feature, with a SHAP value of 5.339. This was followed by Series 1 (5.083), Series 2 (4.595), and Series 3 (3.900). The XGBoost model for female athletes demonstrated the highest predictive performance among all models, with an MAE of $2.027 \pm 0.495$ s and an $R^2$ of $0.986 \pm 0.007$.

Overall, the XGBoost and SHAP results consistently indicated that Series 4 shooting time was the most influential factor affecting total shooting time in both male and female athletes. This finding aligns with the MDI-based results from the Random Forest model, but contrasts with the Permutation Importance analysis for male athletes, which identified Series 1 as the most important feature. Among the three models employed, XGBoost achieved the highest $R^2$ and lowest MAE, suggesting its superior ability to capture and predict the complex relationships within Modern Pentathlon shooting time data.

In this study, we evaluated the predictive performance of Random Forest and XGBoost models by gender (Table 7). The results indicate that, in terms of Mean Absolute Error (MAE), the male group consistently exhibited lower error values compared to the female group, suggesting higher predictive accuracy for male data. Notably, the XGBoost model with SHAP-based feature importance achieved the lowest MAE (1.834) for males, highlighting its superior performance in this subgroup. Conversely, the coefficient of determination ($R^2$) was consistently higher in the female group, with the Random Forest model using MDI achieving the highest $R^2$ value on the test set (0.972), thereby demonstrating stronger explanatory power for female data. These findings suggest that while the models yield more precise predictions for male participants, they better capture the variability in female data. Furthermore, the training $R^2$ values across all models ranged between 0.991 and 0.994, and test $R^2$ values exceeded 0.90, indicating robust model performance with minimal risk of overfitting. Collectively, these results imply that predictive performance may vary by gender, and the choice of model should be aligned with the analytical objective—favoring MAE-oriented models for accuracy-driven tasks and $R^2$-oriented models for interpretability-focused analyses.

**Table 7.** Model performance.

| Model | | Gender | MAE (Mean Absolute Error) | $R^2$ (R-Squared) | |
|-------|--|--------|---------------------------|-------|-------|
| | | | | Train | Test |
| Random Forest | MDI | Male | 1.927 | 0.991 | 0.928 |
| | | Female | 1.975 | 0.993 | 0.972 |
| | Permutation Importance | Male | 1.927 | 0.991 | 0.901 |
| | | Female | 1.975 | 0.994 | 0.965 |
| XGBoost | SHAP | Male | 1.834 | 0.991 | 0.901 |
| | | Female | 2.027 | 0.993 | 0.965 |

## 4. Discussion

This study utilized three machine learning-based variable importance analysis techniques—Mean Decrease Impurity (MDI), Permutation Importance, and SHAP (SHap-

ley Additive exPlanations)—to quantify the relative contribution of individual shooting series' performance to the overall variation in total shooting time.

Prior correlation analysis revealed a significant positive correlation between the shooting time of each series (S1–S4) and the total shooting time. Notably, S4 exhibited the highest correlation coefficient (r). This finding aligns with the machine learning-based importance analysis results (MDI, Permutation Importance, and SHAP), all of which indicated that S4 has the greatest influence on overall shooting performance.

However, it is important to recognize that correlation analysis can only reflect simple linear relationships and is limited in its ability to explain interactions or nonlinear relationships between variables. In contrast, SHAP and Permutation Importance can capture these complex structures, enabling a more detailed assessment of variable importance. Therefore, while correlation analysis is useful for understanding general trends, machine learning-based interpretations that consider individual and nonlinear effects are more suitable for developing practical training strategies.

Based on these results, we will now delve into an in-depth discussion of the findings from the machine learning-based analyses.

### 4.1. Differences in Importance According to Shooting Series Order

The results of this study revealed differences in variable importance based on the Series order in Modern Pentathlon laser run events. First, in all analysis models (Random Forest MDI, XGBoost SHAP), Series 4, the final shooting series, showed the highest importance for both male and female athletes. This suggests that maintaining concentration, rapid recovery ability, and effective psychological pressure control significantly influence final performance due to accumulated fatigue, psychological pressure, and the near-decided competition standings in the latter stages of the event. These findings support previous research indicating that decreased concentration during the latter stages of laser run shooting critically impacts accuracy and competition outcomes [18]. From a sports physiological perspective, this appears to be influenced by fatigue, resistance, and resilience; in other words, the ability to rapidly recover physical and mental stability when transitioning from the 800 m run to shooting, which requires precise movements, is considered a key factor for successful completion of all five shots in each series [19].

Conversely, the Permutation Importance analysis revealed that the shooting time in Series 1 was most important for male athletes, presenting a contrasting result to the previous findings. This suggests that beyond the significance of the later series, the success of early-stage shooting significantly impacts the flow of the competition and psychological stability. From a sports psychology perspective, successful performance in the early stages of a competition can boost self-efficacy and induce a positive psychological state, thereby contributing positively to maintaining performance in subsequent series and running segments [20]. This highlights the importance of early-game psychological strategies, particularly for athletes to effectively manage the anxiety and pressure that can arise during initial shooting. Therefore, to build confidence before the competition and steer the overall flow of the game to their advantage, male athletes should engage in strategic, focused training and mental skills training for both Series 4 and Series 1 [19].

### 4.2. Differences Between Machine Learning Models

Based on the results of the variable importance of the three models utilized in this study, we identify the reasons for the different importance of shooting order across series and discuss their potential use in sport.

The Mean Decrease Impurity (MDI), based on Random Forest, calculates the relative contribution of variables by measuring the decrease in impurity at each node [21]. While it

is known for its fast and intuitive computation, it tends to distort or disperse importance when variables have high correlations, such as in time series data. In this study, for instance, Series 2 and 3 were underestimated in importance for male athletes. This is likely due to the highly correlated nature of shooting times across series in laser run, which is a time-series characteristic. If one primary feature is preferentially selected as a splitting criterion, it creates a structural limitation where other similar, highly correlated variables are underestimated [10]. Consequently, these other correlated variables lose the opportunity to contribute to impurity reduction, leading to an underestimation of their individual importance compared to their actual contribution.

Permutation Importance directly measures the decrease in model performance, offering the advantage of a model-agnostic approach. In this study, Series 1 was identified as a key variable for male athletes, a different finding compared to MDI. This is because Permutation Importance, through its model-agnostic approach [19], directly measures how much the model's actual predictive performance ($R^2$) degrades when specific feature values are randomly shuffled [12]. While this method is less susceptible to importance bias caused by correlations that MDI might exhibit, and more accurately captures the variables the model truly relies on for predictions, it has drawbacks: it is computationally intensive and sensitive to noise [20].

The SHapley Additive exPlanations (SHAP) model, an interpretation technique based on Shapley values from game theory, offers the advantage of fairly and mathematically consistently breaking down and explaining each variable's contribution to predictions [22]. XGBoost, a boosting-based ensemble learning technique, demonstrates excellent predictive performance through its regularization capabilities, ability to detect nonlinear relationships, and variable interactions [23]. SHAP is particularly strong because it can trace back the reasons for a specific prediction within the model, variable by variable, allowing for micro-level analysis tailored to individual athlete performance characteristics [24].

Therefore, the consistent finding in this study that Series 4 had the greatest influence for both male and female athletes suggests that XGBoost accurately identified this variable as a genuinely important predictor. This is not just a simple variable importance measurement; it transforms the contribution of model-based predictions into an interpretable form, effectively reflecting the complex, time-series, and nonlinear nature of sports data. Because SHAP can even visualize contributions in individual predictions, it is considered the most promising method for providing training feedback and practical application in sports competitions.

### 4.3. Research Results-Based Field Utilization Method

According to research, the fourth series significantly impacts shooting performance as competitions progress, primarily because it commences when athletes are fatigued in the latter stages of the event [25]. Therefore, coaches need to focus on training programs for the fourth series that are realistic and enable athletes to maintain high levels of concentration and accuracy even under fatigue. For instance, intermittent high-intensity training, which incorporates short rest intervals during running or high-intensity aerobic exercise alongside shooting practice, can closely simulate actual competition conditions. This approach has the potential to maximize athletes' physical and psychological recovery capabilities. Considering the increased psychological pressure inherent in the fourth series, consistent implementation of shooting routines and ongoing psychological training would likely enhance athletes' psychological stability and confidence.

Meanwhile, for male athletes, a Permutation Importance analysis revealed that the first series holds significant importance. This suggests that initial psychological stability and early performance can influence the overall flow of the competition and affect con-

fidence [26]. Consequently, for athletes who exhibit relative instability in the first series, training programs should be structured to prioritize improving the consistency of early shooting performance. This includes focusing on pre- and post-shooting routines and managing psychological anxiety to maintain stability at the start of a competition.

As a limitation of the study, the analysis was based on data from international athletes competing at the elite level; the pattern that emerges for general athletes may be different.

## 5. Conclusions

This study employed multiple machine learning techniques to comprehensively analyze the relative importance of shooting times across series in the laser run event of the Modern Pentathlon. Specifically, three feature importance assessment methods—Random Forest's Mean Decrease in Impurity (MDI), Permutation Importance, and XGBoost combined with SHAP (SHapley Additive exPlanations)—were applied to evaluate the impact of each shooting series on total shooting time.

First, in MDI using the random forest model, Series 4 (final shooting series) consistently demonstrated the highest importance for both male and female athletes. This result suggests that under accumulated fatigue and extreme mental stress conditions, the ability to maintain concentration, achieve rapid recovery, and effectively manage psychological pressure during the race-ending phase plays a decisive role in overall competitive performance.

Second, the Permutation Importance analysis identified Series 1 as the most influential feature for male athletes. This highlights the substantial significance of early-stage shooting performance in shaping initial psychological stability and overall race dynamics. It further underscores the importance of implementing targeted psychological strategies and focused training regimens for the preliminary phase of competition.

Third, the XGBoost + SHAP (SHAPley Additive ex-Planations) model consistently showed that Series 4 was the most influential factor on total shooting time in both male and female athletes; of the three models used, it had the highest $R^2$ and the lowest MAE.

In conclusion, machine learning analysis was performed based on MDI, Permutation Importance, and SHAP techniques to identify the main points of time that affect the shooting of the Modern Pentathlon. The SHAP-based XGBoost model demonstrated the highest predictive accuracy and interpretability, confirming that Shooting Series 4 is the most decisive variable for both male and female athletes. These results suggest that the ability to maintain psychological and physiological resilience and concentration in the second half of the game affects performance, and a strategic approach that takes this into account is needed when designing a training program.

As a limitation of the study, SHAP can technically be applied to random forest models; however, we did not use it in this study because the computational cost of tree ensembles is significantly higher and their interpretability is lower compared to gradient boosting models such as XGBoost. Therefore, SHAP + Random Forest was not utilized because XGBoost is a more practical option for this analysis, as it provides a more stable structure for calculating SHAP values.

**Author Contributions:** Conceptualization, J.P. and J.K.; methodology, J.P. and J.K.; software, J.K.; validation, J.K.; formal analysis, J.P. and J.K.; investigation, J.P. and J.K.; data curation, J.K.; writing—original draft preparation, J.P. and J.K.; writing—review and editing, J.P. and J.K.; supervision, J.P.; project administration, J.P. All authors have read and agreed to the published version of the manuscript.

**Funding:** This research received no external funding.

**Institutional Review Board Statement:** Analysis of non-human anonymized data.

**Informed Consent Statement:** Not applicable.

**Data Availability Statement:** This research study analyzed publicly available data. https://www.uipmworld.org/past-events/2+5+10+116 (accessed on 10 December 2024).

**Conflicts of Interest:** The authors declare no conflicts of interest.

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
