# Peer review of "Applying Machine Learning for Analyzing Shooting Importance in Modern Pentathlon"

_applsci, doi:10.3390/app15179378_

Round 1

Reviewer 1 Report

Comments and Suggestions for Authors

This paper has some potential but would benefit from rewriting. Below are my itemized comments:

  1. The figures and tables are not always clearly or directly discussed in the main text, which makes them difficult to follow. Adding explicit references and clearer explanations in the text would help readers better understand and interpret the results.
  2. If I understand correctly, this paper describes using machine learning to analyze which shooting round most affects total time in the Modern Pentathlon Laser-Run. However, I have a hard time understanding the goal of the phrase “relative importance of each series' shooting time (Series 1 ~ Series 4) to the total shooting time (Total Series).” This is confusing because the total is simply the sum of the four series times. Please clarify whether you’re actually trying to attribute variation in total time to specific series (i.e., decomposition of variance) rather than predict the total itself.
  3. I’m also having trouble understanding exactly how the prediction models are structured. Are you using each individual series time separately to predict total shooting time (e.g., S1 → Total, S2 → Total), or are you including all four series times together as predictors in the same model? If it’s the latter (all series included at once), it seems inevitable that Series 4 will appear highly important since total time is simply the sum of all series. Could you please clarify how the models handle this relationship and explain why machine learning adds value over simply summing the times or using basic regression?
  4. The paper uses many machine learning methods and jargon (Random Forest, Permutation Importance, SHAP) to analyze what is fundamentally a deterministic sum of four variables. While I understand that feature importance analysis can partition variance contributions, it is not clear that these complex methods add substantial insight beyond what simple descriptive statistics or linear regression could provide. Could the authors please clarify how these machine learning approaches offer a better or more meaningful analysis than, for example, basic correlation analysis between each series and total shooting time?
  5. I would also like to see more clarity on how these findings translate into concrete training recommendations. At a high level of competition, all four shooting series are important, so simply knowing that Series 4 has high importance may not meaningfully change training practices. Could the authors be more specific about how coaches or athletes should use these results to adjust their preparation? For example, what types of drills or psychological strategies would specifically target the issues identified in Series 4?
  6. Consider including simple baseline analyses (e.g., linear regression, correlations) to demonstrate whether the machine learning models meaningfully outperform these approaches. This would strengthen the justification for using more complex methods.
  7. The abstract currently describes Permutation Importance and SHAP as if they are machine learning models. These are interpretability methods applied to trained models. I suggest clarifying this distinction to improve accuracy.
  8. It is a bit unusual to start the abstract with “Background:” and include sections such as “Results:” and “Conclusions:”. Consider reorganizing the abstract to improve readability and flow.
  9. In Figure 1, does “ST” mean “shooting time”? If so, please do not use the abbreviation, as it is not introduced or explained anywhere else.

Author Response

Thank you for your careful examination for the high-quality paper, and I have written and attached the contents of the modifications in a word file.

Reviewer 2 Report

Comments and Suggestions for Authors

The paper investigates the application of machine learning in sports, with a particular emphasis on incorporating explainable artificial intelligence techniques. While the topic is both relevant and timely, several critical issues must be addressed before the manuscript can be considered for publication. These include inadequate referencing of figures and tables, duplicate table numbering, and insufficient methodological detail.

  • The introduction should clearly define XAI and explain its significance in machine learning studies. It should also clarify that ML models are primarily used for prediction or classification tasks, while XAI techniques are intended to interpret or explain these predictions. These approaches are complementary and should be presented as such. Additionally, the paper should include a dedicated section explaining the rationale for selecting specific ML models, highlighting their strengths and limitations in the context of the study.

  • I recommend that the authors emphasize SHAP’s model-agnostic nature and clarify that it can be used to interpret feature importance and model behavior across all the ML models discussed. If SHAP was applied exclusively to XGBoost, this choice should be explicitly justified, and its broader applicability acknowledged.

  • The manuscript lacks sufficient detail regarding the hyperparameter tuning process. For each ML model used, the authors should clearly describe: The specific hyperparameters that were tuned, The range of values explored, The tuning strategy employed, The final values selected for the best-performing models

  • I strongly recommend including a table that presents performance metrics for the training, validation, and testing datasets. This is crucial for assessing model generalization. A large gap between training and test performance, for example, would suggest potential overfitting.

  • All figures and tables must be referenced and discussed within the main text. This is a fundamental requirement for scientific manuscripts. Each figure should be described in terms of what it illustrates, and each table should be accompanied by an explanation of the results it presents.

  • I suggest that the authors consider merging the Results and Discussion sections. In this case, such integration could improve the clarity and flow of the paper by immediately linking the results to their interpretation and implications.

Author Response

(The authors gave the same response as above.)

Reviewer 3 Report

Comments and Suggestions for Authors

The manuscript presents a well-structured and timely investigation into the importance of shooting series in modern pentathlon using machine learning techniques. The study leverages robust methodologies (Random Forest, Permutation Importance, SHAP) and provides valuable insights for sports science. However, there are areas where clarity, methodological rigor, and interpretation could be improved. Below are detailed critiques and suggestions for each section.

  1. Abstract

-The abstract could better emphasize the novelty of the study (e.g., comparison of multiple ML interpretability methods in pentathlon). Explicitly state the gap in prior research (e.g., lack of multi-model interpretability analysis in pentathlon).

-Metrics (MAE, R²) are mentioned but lack context (e.g., how they compare to baseline or prior work). Briefly compare model performances (e.g., "XGBoost+SHAP outperformed others with R²=0.97").

  1. Introduction 

-The rationale for selecting MDI, Permutation Importance, and SHAP is underdeveloped. Why these three? How do they complement each other?

-The introduction could better motivate the study’s uniqueness (e.g., first application of SHAP to pentathlon data).

  1. Materials and Methods

-Data Limitations: No discussion of potential biases (e.g., differences in competition rounds, athlete skill levels).

-Model Tuning: Hyperparameter tuning processes for Random Forest/XGBoost are not described.

-Ethical Considerations: Missing statement on data anonymization/IRB approval (later noted in "Institutional Review Board Statement").

  1. Results

-Inconsistencies: Discrepancies in importance rankings (e.g., Series 1 vs. Series 4 for males) are not thoroughly discussed until the Discussion section.

-Visualizations: Figures lack captions explaining axes/units (e.g., SHAP values in Table 4).

  1. Discussion

-Practical Implications: How should coaches prioritize Series 1 vs. Series 4 training? Needs actionable recommendations.

-Limitations: No discussion of external validity (e.g., generalizability to non-elite athletes).

  1. Conclusions

The section overstates XGBoost+SHAP’s superiority without noting trade-offs (e.g., computational cost). Please balance conclusions with limitations (e.g., "While XGBoost+SHAP offers high interpretability, its complexity may hinder real-time use").

Author Response

(The authors gave the same response as above.)

Round 2

Reviewer 1 Report

Comments and Suggestions for Authors

Overall, the manuscript has improved substantially.

I still think the relative importance of each shooting series to the total time can be estimated using simple correlation or regression analysis. While these approaches might perform somewhat worse than more sophisticated methods, they would serve as a useful baseline.

For example, you have data for four shooting series, s1, s2, s3, and s4, where each represents the shooting time in that series across all observations (e.g., s1 = [t1, t2, t3, ...]). You also have the total time for the four series. The correlation coefficient between each series and the total time (e.g., r1 = corrcoef(s1, total_time), r2 = corrcoef(s2, total_time), etc.) can provide a simple measure of the relative importance of each series.

The authors may consider including such an analysis as a comparison baseline.

Reviewer 2 Report

Comments and Suggestions for Authors

There are several issues not addressed in the previous report, which suggests that the authors did not revise the document carefully. As I explained, Table 4 is never cited in the manuscript. The authors were advised in the previous report that this was a major issue for a scientific publication, yet they failed to correct it. All tables must be cited within the manuscript text. This is a significant concern. Additionally, there are two different tables labeled as Table 2, and another set labeled as Table 3. Tables must be numbered consecutively according to their appearance in the text.

As I also mentioned, Machine Learning (ML) and Explainable AI (XAI) are complementary approaches and should be presented as such. However, this distinction is once again unclear in section 2.4. Lastly, why didn’t you use SHAP with the Random Forest model?

Reviewer 3 Report

Comments and Suggestions for Authors

The manuscript has considered the comments made in the previous review.... I have no further comments from my part
